# Just How Toxic is Data Poisoning? A Benchmark for Backdoor and Data Poisoning Attacks

## Abstract

Data poisoning and backdoor attacks manipulate training data in order to cause models to fail during inference. A recent survey of industry practitioners found that data poisoning is the number one concern among threats ranging from model stealing to adversarial attacks. However, we find that the impressive performance evaluations from data poisoning attacks are, in large part, artifacts of inconsistent experimental design. Moreover, we find that existing poisoning methods have been tested in contrived scenarios, and many fail in more realistic settings. In order to promote fair comparison in future work, we develop standardized benchmarks for data poisoning and backdoor attacks.

## 1 Introduction

*Data poisoning* is a security threat to machine learning systems in which an attacker controls the behavior of a system by manipulating its training data. This class of threats is particularly germane to deep learning systems because they require large amounts of data to train and are therefore often trained (or pre-trained) on large datasets scraped from the web. For example, the Open Images and the Amazon Products datasets contain approximately 9 million and 233 million samples, respectively, that are scraped from a wide range of potentially insecure, and in many cases unknown, sources (Kuznetsova et al., 2020; Ni, 2018). At this scale, it is often infeasible to properly vet content. Furthermore, many practitioners create datasets by harvesting system inputs (e.g., emails received, files uploaded) or scraping user-created content (e.g., profiles, text messages, advertisements) without any mechanisms to bar malicious actors from contributing data. The dependence of industrial AI systems on datasets that are not manually inspected has led to fear that corrupted training data could produce faulty models (Jiang et al., 2017). In fact, a recent survey of 28 industry organizations found that these companies are significantly more afraid of data poisoning than other threats from adversarial machine learning (Kumar et al., 2020).

A spectrum of poisoning attacks exists in the literature. *Backdoor data poisoning* causes a model to misclassify test-time samples that contain a *trigger* – a visual feature in images or a particular character sequence in the natural language setting (Chen et al., 2017; Dai et al., 2019; Saha et al., 2019; Turner et al., 2018). For example, one might tamper with training images so that a vision system fails to identify any person wearing a shirt with the trigger symbol printed on it. In this threat model, the attacker modifies data at both train time (by placing poisons) and at inference time (by inserting the trigger). *Triggerless* poisoning attacks, on the other hand, do not require modification at inference time (Biggio et al., 2012; Huang et al., 2020; Muñoz-González et al., 2017; Shafahi et al., 2018; Zhu et al., 2019; Aghakhani et al., 2020b; Geiping et al., 2020). A variety of innovative backdoor and triggerless poisoning attacks – and defenses – have emerged in recent years, but inconsistent and perfunctory experimentation has rendered performance evaluations and comparisons misleading.

In this paper, we develop a framework for benchmarking and evaluating a wide range of poison attacks on image classifiers. Specifically, we provide a way to compare attack strategies and shed light on the differences between them.

Our goal is to address the following weaknesses in the current literature. First, we observe that the reported success of poisoning attacks in the literature is often dependent on specific (and sometimes unrealistic) choices of network architecture and training protocol, making it difficult to assess the

viability of attacks in real-world scenarios. Second, we find that the percentage of training data that an attacker can modify, the standard budget measure in the poisoning literature, is not a useful metric for comparisons. The flaw in this metric invalidates comparisons because even with a fixed percentage of the dataset poisoned, the success rate of an attack can still be strongly dependent on the dataset size, which is not standardized across experiments to date. Third, we find that some attacks that claim to be "clean label," such that poisoned data still appears natural and properly labeled upon human inspection, are not.

Our proposed benchmarks measure the effectiveness of attacks in standardized scenarios using modern network architectures. We benchmark from-scratch training scenarios and also white-box and black-box transfer learning settings. Also, we constrain poisoned images to be *clean* in the sense of small perturbations. Furthermore, our benchmarks are publicly available as a proving ground for existing and future data poisoning attacks.

The data poisoning literature contains attacks in a variety of settings including image classification, facial recognition, and text classification (Shafahi et al., 2018; Chen et al., 2017; Dai et al., 2019). Attacks on the fairness of models, on on speech recognition, and recommendation engines have also been developed (Solans et al., 2020; Aghakhani et al., 2020a; Li et al., 2016; Fang et al., 2018; Hu et al., 2019; Fang et al., 2020). While we acknowledge the merits of studying poisoning in a range of modalities, our benchmark focuses on image classification since it is by far the most common setting in the existing literature.

## 2    A SYNOPSIS OF TRIGGERLESS AND BACKDOOR DATA POISONING

Early poisoning attacks targeted support vector machines and simple neural networks (Biggio et al., 2012; Koh & Liang, 2017). As poisoning gained popularity, various strategies for triggerless attacks on deep architectures emerged (Muñoz-González et al., 2017; Shafahi et al., 2018; Zhu et al., 2019; Huang et al., 2020; Aghakhani et al., 2020b; Geiping et al., 2020). The early backdoor attacks contained triggers in the poisoned data and in some cases changed the label, thus were not clean-label (Chen et al., 2017; Gu et al., 2017; Liu et al., 2017). However, methods that produce poison examples which don't visibly contain a trigger also show positive results (Chen et al., 2017; Turner et al., 2018; Saha et al., 2019). Poisoning attacks have also precipitated several defense strategies, but sanitization-based defenses may be overwhelmed by some attacks (Koh et al., 2018; Liu et al., 2018; Chacon et al., 2019; Peri et al., 2019).

We focus on attacks that achieve targeted misclassification. That is, under both the triggerless and backdoor threat models, the end goal of an attacker is to cause a target sample to be misclassified as another specified class. Other objectives, such as decreasing overall test accuracy, have been studied, but less work exists on this topic with respect to neural networks (Xiao et al., 2015; Liu et al., 2019). In both triggerless and backdoor data poisoning, the clean images, called *base images*, that are modified by an attacker come from a single class, the *base class*. This class is often chosen to be precisely the same class into which the attacker wants the target image or class to be misclassified.

There are two major differences between triggerless and backdoor threat models in the literature. First and foremost, backdoor attacks alter their targets during inference by adding a trigger. In the works we consider, triggers take the form of small patches added to an image (Turner et al., 2018; Saha et al., 2019). Second, these works on backdoor attacks cause a victim to misclassify any image containing the trigger rather than a particular sample. Triggerless attacks instead cause the victim to misclassify an individual image called the *target image* (Shafahi et al., 2018; Zhu et al., 2019; Aghakhani et al., 2020b; Geiping et al., 2020). This second distinction between the two threat models is not essential; for example, triggerless attacks could be designed to cause the victim to misclassify a collection of images rather than a single target. To be consistent with the literature at large, we focus on triggerless attacks that target individual samples and backdoor attacks that target whole classes of images.

We focus on the *clean-label backdoor attack* and the *hidden trigger backdoor attack*, where poisons are crafted with optimization procedures and do not contain noticeable patches (Saha et al., 2019; Turner et al., 2018). For triggerless attacks, we focus on the *feature collision* and *convex polytope* methods, the most highly cited attacks of the last two years that have appeared at prominent ML conferences (Shafahi et al., 2018; Zhu et al., 2019). We include the recent triggerless methods

*Bullseye Polytope* (BP) and *Witches' Brew* (WiB) in the section where we present metrics on our benchmark problems (Aghakhani et al., 2020b; Geiping et al., 2020). The following section details the attacks that serve as the subjects of our experiments.

**Technical details** Before formally describing various poisoning methods, we begin with notation. Let $X_c$ be the set of all clean training data, and let $X_p = \{x_p^{(j)}\}_{j=1}^J$ denote the set of $J$ poison examples with corresponding clean base image $\{x_b^{(j)}\}_{j=1}^J$. Let $x_t$ be the target image. Labels are denoted by $y$ and $Y$ for a single image and a set of images, respectively, and are indexed to match the data. We use $f$ to denote a feature extractor network.

**Feature Collision (FC)** Poisons in this attack are crafted by adding small perturbations to base images so that their feature representations lie extremely close to that of the target (Shafahi et al., 2018). Formally, each poison is the solution to the following optimization problem.

$$x_p^{(j)} = \underset{x}{\operatorname{argmin}} \|f(x) - f(x_t)\|_2^2 + \beta \|x - x_b^{(j)}\|_2^2. \tag{1}$$

When we enforce $\ell_\infty$-norm constraints, we drop the last term in Equation (1) and instead enforce $\|x_p^{(j)} - x_b^{(j)}\|_\infty \le \varepsilon$, $\forall j$ by projecting onto the $\ell_\infty$ ball after each iteration.

**Convex Polytope (CP)** This attack crafts poisons such that the target's feature representation is a convex combination of the poisons' feature representations by solving the following optimization problem (Zhu et al., 2019).

$$X_p = \underset{\{c_j\},\{x^{(j)}\}}{\operatorname{argmin}} \quad \frac{1}{2} \frac{\|f(x_t) - \sum_{j=1}^J c_j f(x^{(j)})\|_2^2}{\|f(x_t)\|_2^2} \tag{2}$$

$$\text{subject to} \quad \sum_{j=1}^J c_j = 1 \text{ and } c_j \ge 0 \,\forall\, j, \text{ and } \|x^{(j)} - x_b^{(j)}\|_\infty \le \varepsilon \,\forall j$$

**Clean Label Backdoor (CLBD)** This backdoor attack begins by computing an adversarial perturbation to each base image (Turner et al., 2018). Formally,

$$\hat{x}_p^{(j)} = x_b^{(j)} + \underset{\|\delta\|_\infty \le \varepsilon}{\operatorname{argmax}} \mathcal{L}(x_b^{(j)} + \delta, y^{(j)}; \theta), \tag{3}$$

where $\mathcal{L}$ denotes cross-entropy loss. Then, a patch is added to each image in $\{\hat{x}_p^{(j)}\}$ to generate the final poisons $\{x_p^{(j)}\}$. The patched image is subject to an $\ell_\infty$-norm constraint.

**Hidden Trigger Backdoor (HTBD)** A backdoor analogue of the FC attack, where poisons are crafted to remain close to the base images but collide in feature space with a patched image from the target class (Saha et al., 2019). Let $\tilde{x}_t^{(j)}$ denote a patched training image from the target class (this image is not clean), then we solve the following optimization problem to find poison images.

$$x_p^{(j)} = \underset{x}{\operatorname{argmin}} \|f(x) - f(\tilde{x}_t^{(j)})\|_2^2 \text{ s.t. } \|x - x_b^{(j)}\|_\infty \le \varepsilon \tag{4}$$

## 3 WHY DO WE NEED BENCHMARKS?

Backdoor and triggerless attacks have been tested in a wide range of disparate settings. From model architecture to target/base class pairs, the literature is inconsistent. Experiments are also lacking in the breadth of trials performed, sometimes using only one model initialization for all experiments, or testing against one single target image. We find that inconsistencies in experimental settings have a large impact on performance evaluations, and have resulted in comparisons that are difficult to interpret. For example, in CP the authors compare their $\ell_\infty$-constrained attack to FC, which is crafted with an $\ell_2$ penalty. In other words, these methods have never been compared on a level playing field.

To study these attacks thoroughly and rigorously, we employ sampling techniques that allow us to draw conclusions about the attacks taking into account variance across model initializations and class choice. For a single trial, we sample one of ten checkpoints of a given architecture, then randomly select the target image, base class, and base images. In Section 4, all figures are averages from 100 trials with our sampling techniques.

**Disparate evaluation settings from the literature**    To understand how differences in evaluation settings impact results, we re-create the various original performance tests for each of the methods described above within our common evaluation framework. We try to be as faithful as possible to the original works, however we employ our own sampling techniques described above to increase statistical significance. Then, we tweak these experiments one component at a time revealing the fragility of each method to changes in experimental design.

**Establishing baselines**    For the FC setting, following one of the main setups in the original paper, we craft 50 poisons on an AlexNet variant (for details on the specific architecture, see (Krizhevsky et al., 2012; Shafahi et al., 2018)) pre-trained on CIFAR-10 (Krizhevsky et al., 2009), and we use the $\ell_2$-norm penalty version of the attack. We then evaluate poisons on the same AlexNet, using the same CIFAR-10 data to train for 20 more epochs to "fine tune" the model end to end. Note that this is not really transfer learning in the usual sense, as the fine tuning utilizes the same dataset as pre-training, except with poisons inserted (Shafahi et al., 2018).

The CP setting involves crafting 5 poisons using a ResNet-18 model pre-trained on CIFAR-10, and then fine tuning the linear layer of the same ResNet-18 model with a subset of the CIFAR-10 training comprising 50 images per class (including the poisons) (He et al., 2016). This setup is also not representative of typical transfer learning, as the fine-tuning data is sub-sampled from the pre-training dataset. In this baseline we set $\varepsilon = {}^{25.5}/_{255}$ matching the original work (Zhu et al., 2019).

One of the original evaluation settings for CLBD uses 500 poisons. We craft these on an adversarially trained ResNet-18 and modify them with a $3 \times 3$ patch in the lower right-hand corner. The perturbations are bounded with $\varepsilon = {}^{16}/_{255}$. We then train a narrow ResNet model from scratch with the CIFAR-10 training set (including the poisons) (Turner et al., 2018).

For the HTBD setting, we generate 800 poisons with another modified AlexNet (for architectural details, see Appendix A.13) which is pre-trained on CIFAR-10 dataset. Then, an $8 \times 8$ trigger patch is added to the lower right corner of the target image, and the perturbations are bounded with $\varepsilon = {}^{16}/_{255}$. We use the entire CIFAR-10 dataset (including the poisons) to fine tune the last fully connected layer of the same model used for crafting. Once again, the fine-tuning data in this setup is not disjoint from the pre-training data (Saha et al., 2019). See the left-most bars of Figure 3 for all baseline results.

**Inconsistencies in previous work**    The baselines defined above do not serve as a fair comparison across methods, since the original works to which we try and stay faithful are inconsistent. Table 1 summarizes experimental settings in the original works. If a particular component (column header) was considered anywhere in the original paper's experiments, we mark a ($\checkmark$), leaving exes ($\times$) when something was not present in any experiments. Table 1 shows the presence of data normalization and augmentation as well as optimizers (SGD or ADAM). It also shows which learning setup the original works considered: frozen feature extractor (FFE), end-to-end fine tuning (E2E), or from-scratch training (FST), as well as which threat levels were tested, white, grey or black box (WB, GB, BB). We also consider whether or not an ensembled attack was used. The $\varepsilon$ values reported are out of 255 and represent the smallest bound considered in the papers; note FC uses an $\ell_2$ penalty so no bound is enforced despite the attack being called "clean-label" in the original work. We conclude from Table 1 that experimental design in this field is extremely inconsistent.

Table 1: Various experimental designs used in data poisoning research.

| Attack | Data Norm. | Aug. | Opt. SGD | Transfer Learning FFE | E2E | FST | Threat Model WB | GB | BB | Ensembles | $\varepsilon$ |
|---|---|---|---|---|---|---|---|---|---|---|---|
| FC | $\times$ | $\times$ | $\times$ | $\checkmark$ | $\checkmark$ | $\times$ | $\checkmark$ | $\times$ | $\times$ | $\times$ | - |
| CP | $\checkmark$ | $\times$ | $\times$ | $\checkmark$ | $\checkmark$ | $\times$ | $\times$ | $\checkmark$ | $\checkmark$ | $\checkmark$ | 25.5 |
| CLBD | $\times$ | $\checkmark$ | $\checkmark$ | $\times$ | $\times$ | $\checkmark$ | $\times$ | $\times$ | $\checkmark$ | $\times$ | 8 |
| HTBD | $\checkmark$ | $\times$ | $\checkmark$ | $\checkmark$ | $\times$ | $\times$ | $\checkmark$ | $\times$ | $\times$ | $\times$ | 8 |

## 4 JUST HOW TOXIC ARE POISONING METHODS REALLY?

In this section, we look at weaknesses and inconsistencies in existing experimental setups, and how these lead to potentially misleading comparisons between methods. We use our testing framework to put triggerless and backdoor attacks to the test under a variety of circumstances, and get a tighter grip on just how reliable existing poisoning methods are.

**Training without SGD or data augmentation**   Both FC and CP attacks have been tested with victim models pre-trained with the ADAM optimizer. However, SGD with momentum has become the dominant optimizer for training CNNs (Wilson et al., 2017). Interestingly, we find that models trained with SGD are significantly harder to poison, rendering these attacks ineffective in practical settings. Moreover, none of the baselines include simple data augmentation such as horizontal flips and random crops. We find that data augmentation, standard in the deep learning literature, also greatly reduces the effectiveness of all of the attacks. For example, FC and CP success rates plummet in this setting to 51.00% and 19.09%, respectively. Complete results including hyperparameters, success rates, and confidence intervals are reported in Appendix A.3. We conclude that these attacks may be significantly less effective against a real practitioner than originally advertised.

**Victim architecture matters**   Two attacks, FC and HTBD, are originally tested on AlexNet variants, and CLBD is tested with a narrow ResNet. These models are not widely used, and they are unlikely to be employed by a realistic victim. We observe that many attacks are significantly less effective against ResNet-18 victims. See Figure 3, where for example, the success rate of HTBD on these victims is as low as 18%. See Appendix A.4 for a table of numerical results. These ablation studies are conducted in the baseline settings but with a ResNet-18 victim architecture. These ResNet experiments serve as an example of how performance can be highly dependent on the selection of architecture.

**"Clean" attacks are sometimes dirty**   Each of the original works we consider purports to produce "clean-label" poison examples that look like natural images. However these methods often produce easily visible image artifacts and distortions due to the large values of $\epsilon$ used. See Figure 1 for examples generated by two of the methods, where FC perturbs a clean "cat" into an unrecognizable poison (left), and CP generates an extremely noisy poison from a base in the "airplane" class (right). These images are not surprising since the FC method is tested with an $\ell_2$ penalty in the original work, and CP is $\ell_\infty$ constrained with a large radius of $^{25.5}/_{255}$.

In many contexts, avoiding detection by automated systems may be more important than maintaining perceptual similarity. In our work, we focus on perceptual similarity as defined by the $\ell_\infty$ constraint as this reflects the explicit goal of most of the attacks we examine, and it is, in general, a much more common area of study. Adaptive attacks that avoid defense or detection is relatively unexplored and an interesting area for future research (Koh et al., 2018).

Figure 1: Bases (top) and poisons (bottom).

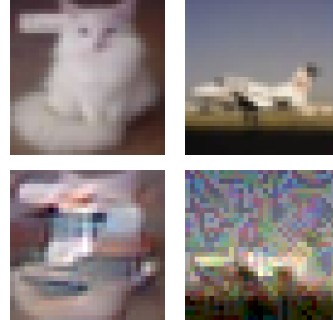

Borrowing from common practice in the evasion attack and defense literature, we test each method with an $\ell_\infty$ constraint of radius $^8/_{255}$ and find that the effectiveness of every attack is significantly diminished (Madry et al., 2017; Dong et al., 2020). Thus, a standardized constraint on poison examples is necessary for fair comparison of attacks, and these previous attacks are not nearly as threatening under constraints that enforce clean poisons. See Figure 3, and see Appendix A.5 for a table of numerical results.

**Proper transfer learning is less vulnerable**   Of the attacks we study here, FC, CP, and HTBD were originally proposed in settings referred to as "transfer learning." Each particular setup varies, but none are true transfer learning since the pre-training datasets and fine-tuning datasets overlap. For example, FC uses the entire CIFAR-10 training dataset for both pre-training and fine tuning. Thus, their threat model entails allowing an adversary to modify the training dataset but only for the last few epochs. Furthermore, these attacks use inconsistently sized fine-tuning datasets.

To simulate transfer learning, we test each attack with ResNet-18 feature extractors pre-trained on CIFAR-100. We fine tune with CIFAR-10 data in both cases, showing that these methods actually perform better in the setting with real transfer learning, *i.e.* where the pre-training data and fine-tuning data are not from the same datasets and do not contain the same classes. In Figure 3, every attack aside from CP shows worse performance when transfer learning is done on data that is disjoint from the pre-training dataset. The attacks designed for transfer learning may not work as advertised in more realistic transfer learning settings. See Appendix A.6.

**Performance is not invariant to dataset size**   Existing work on data poisoning measures an attacker's budget in terms of what percentage of the training data they may modify. This begs the question whether percentage alone is enough to characterize the budget. Does the actual size of the training set matter? We find the number of images in the training set has a large impact on attack performance, and that performance curves for FC and CP intersect. When we hold the percentage poisoned constant at 1%, but we change the number of poisons and the size of the training set accordingly, we see no consistent trends in how the attacks are affected. Figure 2 shows the success of each attack as a function of dataset size (shaded region is one standard error). This observation suggests that one cannot compare attacks tested on different sized datasets by only fixing the percent of the dataset poisoned. See Appendix A.7.

**Black-box performance is low**   Whether considering transfer learning or training from scratch, testing these methods against a black-box victim is surely one of the most realistic tests of the threat they pose. Since, FC, CP and HTBD do not consider the black-box scenario in the original works, we take the poisons crafted using baseline methods and evaluate them on models of different architectures than those used for crafting. The attacks show much lower performance in the black-box settings than in the baselines, in particular FC, CP, and HTBD all have success rates lower than 20%. See Figure 3, and see Appendix A.8 for more details.

Figure 2: Scaling the dataset size while fixing the poison budget.

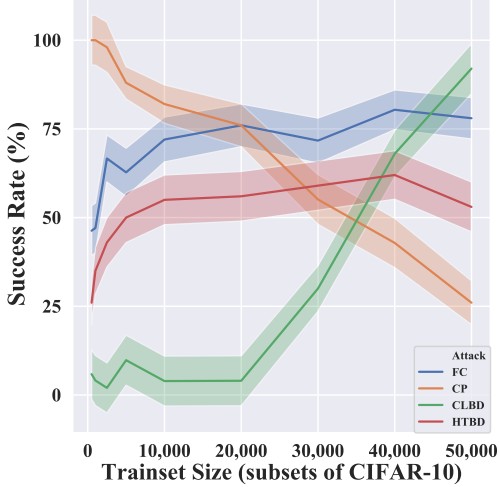

**Small sample sizes and non-random targets**
On top of inconsistencies in experimental setups, existing work on data poisoning often test only on specific target/base class pairs. For example, FC largely uses "frog" as the base class and "airplane" as the target class. CP, on the other hand, only uses "ship" and "frog" as the base and target classes, respectively. Neither work contains experiments where each trial consists of a randomly selected target/base class pair. We find that the success rates are highly class pair dependent and change dramatically under random class pair sampling. Thus, random sampling is critical for performance evaluation. See Appendix A.9 for a comparison of the specific class pairs from these original works with randomly sampled class pairs.

In addition to inconsistent class pairs, data poisoning papers often evaluate performance with very few trials since the methods are computationally expensive. In their original works, FC and CP use 30 and 50 trials, respectively, for each experiment, and these experiments are performed on the same exact pre-trained models each time. And while HTBD does test randomized pairs, they only show results for ten trials on CIFAR-10. These small sample sizes yield wide error bars in performance evaluation. We choose to run 100 trials per experiment in our own work. While we acknowledge that a larger number would be even more compelling, 100 is a compromise between thorough experimentation and practicality since each trial requires re-training a classifier.

**Attacks are highly specific to the target image**   Triggerless attacks have been proposed as a threat against systems deployed in the physical world. For example, blue Toyota sedans may go undetected by a poisoned system so that an attacker may fly under the radar. However, triggerless attacks are generally crafted against a specific target image, while a physical object may appear differently under

difference real-world circumstances. We upper-bound the robustness of poison attacks by applying simple horizontal flips to the target images, and we find that poisoning methods are significantly less successful when the exact target image is unknown. For example, FC is only successful 7% of the time when simply flipping the target image. See Figure 3 and Appendix A.10.

**Backdoor success depends on patch size**    Backdoor attacks add a patch to target images to trigger misclassification. In real-world scenarios, a small patch may be critical to avoid being caught. The original HTBD attack uses an $8 \times 8$ patch, while the CLBD attack originally uses a $3 \times 3$ patch (Saha et al., 2019; Turner et al., 2018). In order to understand the impact on attack performance, we test different patch sizes. We find a strong correlation between the patch size and attack performance, see Appendix A.12. We conclude that backdoor attacks must be compared using identical patch sizes.

Figure 3: We show the fragility of poisoning methods to experimental design. This figure depicts baselines along with the results of ablation studies. Different methods respond differently to these testing scenarios, supporting the need for consistent and thorough testing. Horizontal lines denote performance on baselines described in Section 3, and bars represent the results of changing a specific feature in an individual method's baseline. Tables of these results with confidence intervals can be found in the appendices.

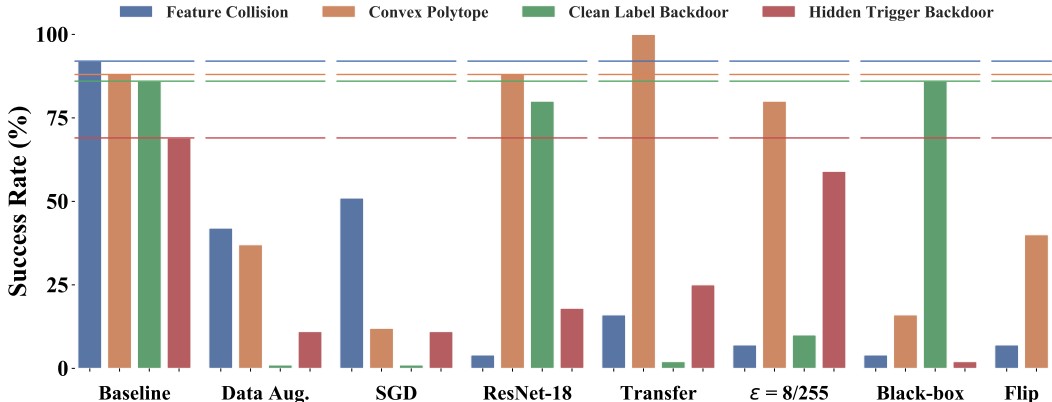

## 5    UNIFIED BENCHMARKS FOR DATA POISONING ATTACKS

**Our Benchmark**    We propose new benchmarks for measuring the efficacy of **both** backdoor and triggerless data poisoning attacks. We standardize the datasets and problem settings for our benchmarks below.[1] Target and base images are chosen from the testing and training sets, respectively, according to a seeded/reproducible random assignment. Poison examples crafted from the bases must remain within the $\ell_\infty$-ball of radius $8/255$ centered at the corresponding base images. Seeding the random assignment allows us to test against a significant number of different random choices of base/target, while always using the same choices for each method, thus removing a source of variation from the results. We consider two different training modes:

I **Transfer Learning:** A feature extractor pre-trained on clean data is frozen and used while training a linear classification head on a disjoint set of training data that contains poisons.

II **Training From Scratch:** A network is trained from random initialization on data containing poison examples in the training set.

To further standardize these tests, we provide pre-trained architectures to test against. The parameters of one model are given to the attacker. We then evaluate the strength of the attacks in white-box and black-box scenarios. For white-box tests in the transfer learning benchmarks, we use the same frozen feature extractor that is given to the attacker for evaluation. While in the black-box setting, we craft poisons using the known model but we test on the two models the attacker has not seen, averaging the results. When training from scratch, models are trained from a random initialization on the poisoned

---

[1]Code is available at (suppressed for anonymity).

dataset. We report averages over 100 independent trials for each test. Backdoor attacks can use any $5 \times 5$ patch. Note that the number of attacker-victim network pairs is kept small in our benchmark because each of the 100 trials requires re-training (in some cases from scratch), and we want to keep the benchmark within reach for researchers with modest computing resources.

**CIFAR-10 benchmarks**  Models are pretrained on CIFAR-100, and the fine-tuning data is a subset of CIFAR-10. We choose this subset to be the first 250 images from each class, allowing for 25 poison examples. This amount of data motivates the use of transfer learning, since training from scratch on only 2,500 images yields poor generalization. See Appendix A.13 for examples. We allow 500 poisons when training from scratch, see Appendix A.15 for a case-study in which we investigate how many poisons an attacker may be able to place in a dataset compiled by querying the internet for images. We allow the attacker access to a ResNet-18, and we do black-box tests on a VGG11 (Simonyan & Zisserman, 2014), and a MobileNetV2 (Sandler et al., 2018), and we use one of each model when training from scratch and report the average.

**TinyImageNet benchmarks**  Additionally, we pre-train VGG16, ResNet-34, MobileNetV2 models on the first 100 classes of the TinyImageNet dataset (Le & Yang, 2015). We fine tune these models on the second half of the dataset, allowing for 250 poison images. As above, the attacker has access to the particular VGG16 model, and black-box tests are done on the other two models. In the from-scratch setting, we train a VGG16 model on the entire TinyImageNet dataset with 250 images poisoned.[2]

**Benchmark hyperparameters**  We pre-train models on CIFAR-100 with SGD for 400 epochs starting with a learning rate of 0.1, which decays by a factor of 10 after epochs 200, 300, and 350. Models pre-trained on the first half of TinyImageNet are trained with SGD for 200 epochs starting with a learning rate of 0.1, which decays by a factor of 10 after epochs 100 and 150. In both cases, we apply per-channel data normalization, random crops, and horizontal flips, and we use batches of 128 images (augmentation is also applied to the poisoned images). We then fine tune with poisoned data for 40 epochs with a learning rate that starts at 0.01 and drops to 0.001 after the 30th epoch (this applies to the transfer learning settings).

When training from scratch on CIFAR-10, we include the 500 perturbed poisons in the standard training set. We use SGD and train for 200 epochs with batches of 128 images and an initial learning rate of 0.1 that decays by a factor of 10 after epochs 100 and 150. Here too, we use data normalization and augmentation as described above. When training from scratch on TinyImageNet, we allow for 250 poisoned images. All other hyperparameters are identical.

Our evaluations of six different attacks are shown in Table 2. These attacks are not easily ranked, as the strongest attacks in some settings are not the strongest in others. Witches' Brew (WiB) is not evaluated in the transfer learning settings, since it is not considered in the original work (Geiping et al., 2020).) See Appendix A.16 for tables with confidence intervals. We find that by using disjoint and standardized datasets for transfer learning, and common training practices like data normalization and scheduled learning rate decay, we overcome the deficits in previous work. Our benchmarks can provide useful evaluations of data poisoning methods and meaningful comparisons between them.

Table 2: Benchmark success rates (%) (best in each column is in bold).

| | CIFAR-10 | | | TinyImageNet | | |
|---|---|---|---|---|---|---|
| | Transfer | | From Scratch | Transfer | | From Scratch |
| Attack | WB | BB | | WB | BB | |
| FC | 22.0 | 7.0 | 1.33 | 49.0 | 2.0 | 4.0 |
| CP | 33.0 | 7.0 | 0.67 | 14.0 | 1.0 | 0.0 |
| BP | **85.0** | 8.5 | 2.33 | **100.0** | **10.5** | **44.0** |
| WiB | - | - | **26.0** | - | - | 32.0 |
| CLBD | 5.0 | 6.5 | 1.00 | 3.0 | 1.0 | 0.0 |
| HTBD | 10.0 | **9.5** | 2.67 | 3.0 | 0.5 | 0.0 |

---

[2]The TinyImageNet from-scratch benchmark is done with 25 independent trials to keep this problem within reach for researchers with modest resources.

## 6  CONCLUSION

The threat of data poisoning is at the forefront of fears around emerging ML systems (Siva Kumar et al., 2020). While many of the methods claiming to do so do not pose a practical threat, some of the recent methods are cause for practitioner concern. With real threats emerging, there is a need for fair comparison. The diversity of attacks, and in particular the difficulty in ordering them by efficacy, calls for a diverse set of benchmarks. With those we present here, practitioners and researchers can compare attacks on a level playing field and gain an understanding of how existing methods match up with one another and where they might fail.

Since the future advancement of these methods is inevitable, our benchmarks will also serve the data poisoning community as a standardized test problem on which to evaluate and future attack methodologies. As even stronger attacks emerge, trepidation on the part of practitioners will be matched by the potential harm of poisoning attacks. We are arming the community with the high quality metrics this evolving situation calls for.

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

## A APPENDIX

### A.1 TECHNICAL SETUP

We report confidence intervals of radius one standard error, $\mathcal{E} = \sqrt{\hat{p}(1-\hat{p})/N}$, where $\hat{p}$ is the observed probability of success, and $N$ is the number of trials. If there are fewer than five observed successes or failures, we set $\hat{p} = 1/2$ to upper-bound standard error.

**Hyperparameters** We use one of seven sets of hyperparameters when training models. We refer to these by name throughout this appendix, and Table 3 shows each setup. For all models trained with SGD, we set the momentum coefficient to 0.9. We always use batches of 128 images and weight decay with a coefficient of $2 \times 10^{-4}$. The "Decay Schedule" column details the epochs after which the learning rate drops by the corresponding decay factor.

Table 3: Hyperparameters.

|        | Learning rate | | | | |
| --- | --- | --- | --- | --- | --- |
| Setup | Initial | Decay Factor | Decay Schedule | Epochs | Optimizer |
| A | 0.001 | 0.5 | 32, 64, 96, 128, 160, 192 | 200 | ADAM |
| B | 0.010 | 0.1 | 100, 150 | 200 | SGD |
| C | 0.100 | 0.1 | 100, 150 | 200 | SGD |
| D | 0.100 | 0.1 | 200, 300, 350 | 400 | SGD |
| E | 0.100 | 0.1 | 40, 60 | 100 | SGD |
| F | 0.100 | 0.1 | 75, 90 | 100 | SGD |
| G | 0.010 | 0.1 | 30 | 40 | SGD |

### A.2 BASELINES

Table 4 shows the baseline performance of each attack. This table reports averages over 100 independent trials with confidence intervals of width one standard error. The experimental setups are summarized in Section 3, and we report additional details here. When we say that an experiment uses a particular architecture, we mean that each trial randomly selects one of ten pre-trained models of this type. The average performances for these sets of pre-trained models are reported in Table 16 below where the hyperparameters and training routines are detailed.

**Feature Collision** The FC baseline uses an AlexNet variant without data normalization or data augmentation. We use the unconstrained version of this attack with the $\ell_2$ penalty in the optimization problem. The algorithm presented in the original work has hyperparameters which we set as follows (Shafahi et al., 2018). We add a watermark of the target image with 30% opacity to each base before the optimization and we use a step size of 0.0001 with the maximum number of iterations set to 1,200. When fine tuning on the poisoned data, we train for 20 epochs with ADAM and a fixed learning rate of $0.001 \times 0.5^6 = 0.00015625$, which is the smallest learning rate used when pre-training.

**Convex Polytope** The CP baseline uses a ResNet-18 with data normalization (without data augmentation). In the poison crafting procedure, we use the ADAM optimizer with a learning rate of 0.04 for a maximum of 4,000 iterations or when the CP loss is less than or equal to $1 \times 10^{-6}$. We bound the perturbations with $\varepsilon = 25.5/255$. Then, we fine tune the model with ADAM for 10 epochs on the poisoned data with a learning rate of 0.1.

**Clean Label Backdoor** The CLBD baseline is a training from scratch scenario. The model used for crafting is an adversarially trained ResNet-18, and we use 20-step PGD with a step size of $4/255$ and $\varepsilon$ of $16/255$ to compute the adversarial perturbations (Madry et al., 2017). Then we train a narrow ResNet model (see (Turner et al., 2018) for architectural details) from scratch using hyperparameter set E as defined in Table 3.

**Hidden Trigger Backdoor**    The HTBD baseline uses a modified AlexNet model with data normalization. Poisons are crafted using SGD for a maximum of 5,000 iterations with initial learning rate of 0.001, which decays by a factor of 0.95 every 2,000 iterations with $\epsilon = {}^{16}/_{255}$. The target image is patched with an $8 \times 8$ patch in the bottom right corner. Then we fine-tune the last linear layer of the network using SGD for 20 epochs with initial learning rate of 0.5, which decays by a factor of 0.1 after epochs 5, 10, and 15.

Table 4: Baseline performance.

| Attack | Success Rate (%) |
|---|---|
| FC | $92.00 \pm 2.71$ |
| CP | $88.00 \pm 3.25$ |
| CLBD | $86.00 \pm 3.47$ |
| HTBD | $69.00 \pm 4.62$ |

## A.3    TRAINING WITHOUT SGD OR DATA AUGMENTATION

We add data normalization and augmentation to the pre-training processes in each attack. For FC and CP, which were originally tested with ADAM, we show results from experiments where normalization and augmentation are used with ADAM as well as when we pre-train with SGD.

Table 5: Data normalization and augmentation + ADAM.

| Attack | Success Rate (%) | Diff. From Baseline (%) |
|---|---|---|
| FC | $42.00 \pm 4.94$ | $-50.00$ |
| CP | $37.00 \pm 4.83$ | $-51.00$ |

Table 6: Data normalization and augmentation + SGD.

| Attack | Success Rate (%) | Diff. From Baseline (%) |
|---|---|---|
| FC | $51.00 \pm 5.00$ | $-41.00$ |
| CP | $12.00 \pm 3.25$ | $-76.00$ |
| CLBD | $1.00 \pm 5.00$ | $-85.00$ |
| HTBD | $11.00 \pm 3.13$ | $-58.00$ |

## A.4    VICTIM ARCHITECTURE MATTERS

We test each method on ResNet-18 victims. Note that CP shows no change from the baseline, as our baseline set-up for CP uses a ResNet-18 victim model.

Table 7: ResNet-18 victims.

| Attack | Success Rate (%) | Diff. From Baseline (%) |
|---|---|---|
| FC | $4.00 \pm 5.00$ | $-88.00$ |
| CP | $88.00 \pm 3.25$ | $0.00$ |
| CLBD | $80.00 \pm 4.00$ | $-6.00$ |
| HTBD | $18.00 \pm 3.84$ | $-51.00$ |

## A.5    "CLEAN" ATTACKS ARE SOMETIMES DIRTY

We test each attack with an $\ell_\infty$-norm constrained perturbation with $\varepsilon = {}^{8}/_{255}$. Note that HTBD shows no change form the baseline since this was the $\varepsilon$ values used in our baseline for this attack. See Table 8.

Table 8: Poisons crafted with $\varepsilon = 8/255$.

| Attack | Success Rate (%) | Diff. From Baseline (%) |
|---|---|---|
| FC | $7.00 \pm 2.55$ | $-85.00$ |
| CP | $80.00 \pm 4.00$ | $-8.00$ |
| CLBD | $10.00 \pm 3.00$ | $-76.00$ |
| HTBD | $59.00 \pm 4.96$ | $-10.00$ |

## A.6 PROPERLY TRANSFER LEARNED MODELS ARE VULNERABLE

We use feature extractors pre-trained to classify CIFAR-100 data to craft the poisons. Then, we use those same feature extractors in the fine-tuning stage when we train the models to classify CIFAR-10 data. See Table 9.

Table 9: Transfer learned victims.

| Attack | Success Rate (%) | Diff. From Baseline (%) |
|---|---|---|
| FC | $16.00 \pm 3.67$ | $-76.00$ |
| CP | $100.00 \pm 5.00$ | $+12.00$ |
| CLBD | $2.00 \pm 5.00$ | $-84.00$ |
| HTBD | $25.00 \pm 4.33$ | $-44.00$ |

## A.7 PERFORMANCE IS NOT INVARIANT TO DATASET SIZE

We study the effect of scaling the dataset size while holding the percentage of data that is poisoned constant. We test each attack with 5 poisons and 500 training images and increment both the number of poison examples and the training set size until we reach 500 poisons and 50,000 training images (the entire CIFAR-10 training set). For every training set size, we allow the attacker to perturb 1% of the data and we see that the strength of poisoning attacks does not scale with any generality – in some cases we see success rates drop with increase in dataset size, and some attacks are more successful with more data. See Table 10 for complete numerical results with confidence intervals of width one standard error.

In addition, we investigate these dynamics with exactly the CIFAR-10 transfer learning benchmark set up. This way, we can evaluate each attack in exactly the same setting, as opposed to above, where we use respective baseline setups. Figure 4 shows that when tested in the exact same evaluation setting, these attacks scale differently with the size of the dataset. Table 11 shows complete numerical results with confidence intervals of width one standard error. This experiment, whose results are perhaps best presented in Figure 2, shows that discussing the poison budget only as a percentage of the data does not allow for fair comparison.

## A.8 BLACK-BOX PERFORMANCE IS LOW

When tested in the black-box setting, all methods except for CLBD show dramatically lower performance. CLBD is intended for use in the training from scratch case, and the particular architectures for crafting and testing are different. So for this experiment, we consider the CLBD baseline black-box. For FC, CP, and HTBD we craft poisons on the architectures used in the baselines. The black-box victims for FC and HTBD are ResNet-18 models, whereas the CP baseline used a ResNet-18 victim, so we use a MobileNetV2 for the black-box victim. See Table 12.

## A.9 SMALL SAMPLE SIZES AND NON-RANDOM TARGETS

We test FC and CP with the specific target/base class pairs studied in the original works. We find the performance of each attack measured only on these classes differs from our baseline. See Table 13. This fact alone is sufficient evidence that the comparisons done in the poison literature are lacking consistency, and that this field needs a benchmark problem.

Table 10: Success rates (%) with varying dataset sizes and number of poisons.

| | Number of Poisons | | | | |
|---|---|---|---|---|---|
| Attack | 5 | 10 | 25 | 50 | 100 |
| FC | $46.30 \pm 6.79$ | $47.06 \pm 6.99$ | $66.67 \pm 6.60$ | $62.75 \pm 6.77$ | $72.00 \pm 6.35$ |
| CP | $100.00 \pm 7.07$ | $100.00 \pm 7.07$ | $98.00 \pm 7.07$ | $88.00 \pm 4.60$ | $82.00 \pm 5.43$ |
| CLBD | $5.88 \pm 7.00$ | $4.08 \pm 7.14$ | $2.00 \pm 7.07$ | $9.80 \pm 7.00$ | $3.92 \pm 7.00$ |
| HTBD | $26.00 \pm 4.39$ | $35.00 \pm 4.77$ | $43.00 \pm 4.95$ | $50.00 \pm 5.00$ | $55.00 \pm 4.97$ |

| | Number of Poisons | | | |
|---|---|---|---|---|
| Attack | 200 | 300 | 400 | 500 |
| FC | $76.00 \pm 6.04$ | $71.70 \pm 6.19$ | $80.39 \pm 5.56$ | $78.00 \pm 5.86$ |
| CP | $76.00 \pm 6.04$ | $55.10 \pm 7.11$ | $42.86 \pm 7.07$ | $26.00 \pm 6.20$ |
| CLBD | $4.00 \pm 7.07$ | $30.00 \pm 6.48$ | $68.00 \pm 6.60$ | $92.00 \pm 7.07$ |
| HTBD | $56.00 \pm 4.96$ | $59.00 \pm 4.92$ | $62.00 \pm 4.85$ | $53.00 \pm 4.99$ |

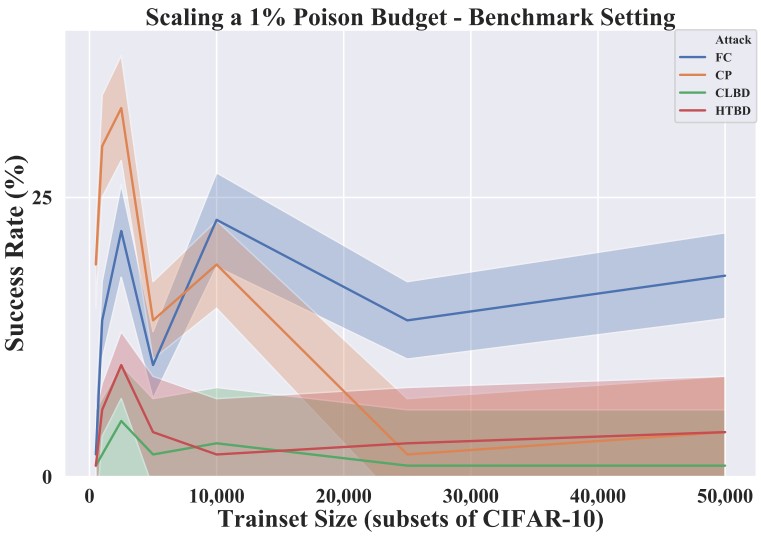

Figure 4: Scaling the dataset size in the benchmark setting.

## A.10 ATTACKS ARE HIGHLY SPECIFIC TO THE TARGET IMAGE

We consider the case where the target object is photographed in a slightly different environment than in the particular image the attacker uses while crafting poisons. Perhaps, the attacker is trying to keep their own red car from being classified as a car. In reality, the deployed model may see a different image than the specific photograph to which the attacker has access. We are unable to get new photographs of the exact objects in CIFAR-10 images, so we choose to upper bound performance on highly modified images by simply flipping the target images horizontally during evaluation. In this setting, we observe that triggerless attacks are severely impaired, supporting our conclusion that they pose less practical threat in physical settings than suggested in previous work. See Table 14.

Table 11: Success rates (%) with varying dataset size in the benchmark setting.

| Attack | Number of Poisons | | | |
|--------|-------|-------|-------|-------|
|        | 5 | 10 | 25 | 50 |
| FC | $2.30 \pm 5.00$ | $14.0 \pm 3.47$ | $22.0 \pm 4.14$ | $10.0 \pm 3.00$ |
| CP | $19.0 \pm 3.92$ | $29.0 \pm 4.61$ | $33.0 \pm 4.70$ | $19.0 \pm 3.92$ |
| CLBD | $1.0 \pm 5.00$ | $2.0 \pm 5.00$ | $5.0 \pm 5.00$ | $2.0 \pm 5.00$ |
| HTBD | $1.0 \pm 5.00$ | $6.0 \pm 2.37$ | $10.0 \pm 3.00$ | $4.0 \pm 5.00$ |

| Attack | Number of Poisons | | |
|--------|------|------|------|
|        | 100 | 250 | 500 |
| FC | $23.0 \pm 4.21$ | $14.0 \pm 3.47$ | $18.0 \pm 3.84$ |
| CP | $19.0 \pm 3.92$ | $2.0 \pm 5.00$ | $4.0 \pm 5.00$ |
| CLBD | $3.0 \pm 5.00$ | $1.0 \pm 5.00$ | $1.0 \pm 5.00$ |
| HTBD | $2.0 \pm 5.00$ | $3.0 \pm 5.00$ | $4.0 \pm 5.00$ |

Table 12: Black-box victim.

| Attack | Success Rate (%) | Diff. From Baseline (%) |
|--------|------------------|-------------------------|
| FC | $4.00 \pm 5.00$ | $-88.00$ |
| CP | $16.00 \pm 3.67$ | $-72.00$ |
| CLBD | $86.00 \pm 3.47$ | $0.00$ |
| HTBD | $2.00 \pm 5.00$ | $-67.00$ |

Table 13: Success with specific class pairs.

| Attack | Target | Base | Success Rate (%) | Diff. From Baseline (%) |
|--------|--------|------|------------------|-------------------------|
| FC | plane | frog | $80.00 \pm 4.00$ | $-12.00$ |
| CP | frog | ship | $83.00 \pm 3.76$ | $-5.00$ |

Table 14: Success on flipped targets.

| Attack | Success Rate (%) | Diff. From Baseline (%) |
|--------|------------------|-------------------------|
| FC | $7.00 \pm 2.55$ | $-85.00$ |
| CP | $40.00 \pm 4.90$ | $-48.00$ |

## A.11 Ensemblizing boosts the attacker

We study the impact of ensemblizing attacks, where the attacker uses several architectures while crafting poisons. This was suggested and tested with CP in the original work (Zhu et al., 2019). In that study however, the comparison between CP and FC is incomplete. We show that ensemblizing helps both attacks and that FC outperforms CP in the white-box setting with enough poisons (both in the single model and the ensemblized attacks). See he rows in Table 22 corresponding to the ensembled FC attack (FC-Ens.) and the ensembled CP attack (CP-Ens.).

## A.12 Backdoor success depends on patch size

In order to determine the effect of the particular size of the patch used in backdoor attacks, we test the backdoor methods with a variety of patch sizes. We see a strong correlation between patch size and success rate. See Table 15. Note that dashes correspond to an attack's baseline performance, see Table 4.

Table 15: Success rates (%) of backdoor attacks with varying patch sizes.

| Attack | Patch Size | | |
|---|---|---|---|
| | $3 \times 3$ | $5 \times 5$ | $8 \times 8$ |
| HTBD | $20.00 \pm 4.0$ | $33.00 \pm 4.70$ | - |
| CLBD | - | $97.00 \pm 5.00$ | $100 \pm 5.0$ |

A.13  MODEL TRAINING AND PERFORMANCES

**Models trained for our experiments**  In Tables 16 - 19, we show the training setups, including references to sets of hyperparameters outlined in Table 3, and the training and testing accuracy of the models we use in this study. Each row in these two tables shows averages of ten models we trained from random intializations with identical training setups. Note that the models called "AlexNet" are the variants introduced in the original FC paper, see that work for details (Shafahi et al., 2018). Models named "HTBD AlexNet" are the modified AlexNet architecture we used for the HTBD experiments and the details are below.

**Architectures we use**  Four of the five architectures we use in this study are widely used and/or detailed in other works. For the modified AlexNet used in FC experiments, see (Shafahi et al., 2018). For ResNet-18 architecture details, see (He et al., 2016). For MobileNetV2, see (Sandler et al., 2018). For VGG11, see (Simonyan & Zisserman, 2014).

**HTBD AlexNet**  The model used in the original HTBD work was a simplified version of AlexNet. But for our baseline experiments we adapt the ImageNet AlexNet model to CIFAR-10 dataset. We modify the kernel size and stride in the first convolution layer to 3 and 2, respectively, in order to take $32 \times 32 \times 3$ input images. For deeper layers we use a stride of 1. The width of the network at then end of the convolutional layers is 256.

Table 16: CIFAR-10 models.

| Model | Norm. | Aug. | Hyperparam. | Train Acc. (%) | Test Acc. (%) |
|---|---|---|---|---|---|
| AlexNet | ✗ | ✗ | A | 99.99 | 73.96 |
| AlexNet | ✓ | ✗ | A | 99.99 | 74.45 |
| AlexNet | ✓ | ✓ | A | 90.35 | 82.36 |
| AlexNet | ✓ | ✓ | B | 98.77 | 85.91 |
| HTBD AlexNet | ✓ | ✗ | B | 100.00 | 77.35 |
| HTBD AlexNet | ✓ | ✓ | B | 98.80 | 84.30 |
| ResNet-18 | ✗ | ✗ | C | 100.00 | 87.05 |
| ResNet-18 | ✓ | ✗ | C | 100.00 | 87.10 |
| ResNet-18 | ✓ | ✓ | C | 99.99 | 94.96 |
| MobileNetV2 | ✗ | ✗ | C | 99.99 | 82.11 |
| MobileNetV2 | ✓ | ✗ | C | 99.99 | 82.04 |
| MobileNetV2 | ✓ | ✓ | C | 99.88 | 93.36 |

Table 17: CIFAR-100 models.

| Model | Norm. | Aug. | Hyperparam. | Train Acc. (%) | Test Acc. (%) |
|---|---|---|---|---|---|
| ResNet-18 | ✓ | ✓ | D | 99.97 | 74.37 |
| MobileNetV2 | ✓ | ✓ | D | 99.95 | 71.81 |
| VGG11 | ✓ | ✓ | D | 99.97 | 67.87 |

**Transfer learning**  We also train models of each architecture from scratch on the first 250 images per class of CIFAR-10. By comparing these models to transfer learned models on the same data, we see the benefit of transfer learning for this quantity of data. Each row of Table 20 shows averages of

Table 18: TinyImageNet models (first 100 classes).

| Model | Norm. | Aug. | Hyperparam. | Train Acc. (%) | Test Acc. (%) |
|---|---|---|---|---|---|
| VGG16 | ✓ | ✓ | C | 99.99 | 63.12 |
| MobileNetV2 | ✓ | ✓ | C | 99.99 | 67.16 |
| ResNet-34 | ✓ | ✓ | C | 99.99 | 62.11 |

Table 19: TinyImageNet models (all classes).

| Model | Norm. | Aug. | Hyperparam. | Train Acc. (%) | Test Acc. (%) |
|---|---|---|---|---|---|
| VGG16 | ✓ | ✓ | C | 99.98 | 58.72 |
| ResNet-34 | ✓ | ✓ | C | 99.98 | 58.08 |

10 models. For the transfer learned models we use exactly the feature extractors from the benchmark, and the pre-trained models' performances are in Table 17. It is clear from Table 20 that with so little data, training from scratch leads to less-than-optimal test accuracy. This motivates the transfer learning benchmark tests since transfer learning does improve performance in this setting.

Table 20: CIFAR-10 models with Trainset size 2500.

| | Model | Norm. | Aug. | Hyperparam. | Train Acc. (%) | Test Acc. (%) |
|---|---|---|---|---|---|---|
| From Scratch | ResNet-18 | ✓ | ✓ | F | 99.12 | 59.71 |
| | MobileNetV2 | ✓ | ✓ | F | 98.99 | 68.55 |
| | VGG11 | ✓ | ✓ | C | 97.98 | 60.72 |
| Transfer learned | ResNet-18 | ✓ | ✓ | G | 99.98 | 80.38 |
| | MobileNetV2 | ✓ | ✓ | G | 99.97 | 79.53 |
| | VGG11 | ✓ | ✓ | G | 99.93 | 74.95 |

## A.14 ADDITIONAL EXPERIMENTS

**Backdoor triggers** Different backdoor attacks use different patch images as a trigger. Does the performance of the attack depends on the patch used? In order to study the impact, we swap the patches of CLBD and HTBD. We resize the CLBD patch to $8 \times 8$ and the HTBD patch to $3 \times 3$ which are the baseline sizes, see Figure 5. We observe that patch image does matter and it can have a significant effect on the performance of an attack. See Table 21.

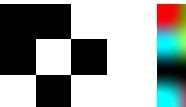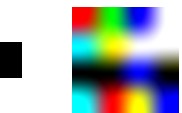

Figure 5: CLBD patch (left), and HTBD patch (right).

Table 21: Success rates (%) of backdoor attacks with swapped patch images.

| Attack | Success Rate (%) | Diff. From Baseline (%) |
|---|---|---|
| HTBD w/ CLBD patch | $51.00 \pm 4.99$ | -8.00 |
| CLBD w/ HTBD patch | $2.00 \pm 5.00$ | -84.00 |

**Poison budget** We conduct an additional experiment to assess the impact of the budget in our benchmark. We test each attack in the same setting at the benchmark, where we do 100 trials each with 50, 100, and 250 poisons, holding the dataset size constant. See Table 22. We see the expected rise in success rate with increased budget, however we note that these increases are almost always

small. We choose to use 25 poisons in the benchmark tests for the following two reasons. First, we want the evaluation of an attack to be accessible to those with modest computing resources. Second, as discussed in Appendix A.15 we find 25 images out of 250 images per class to be a large budget in realistic settings.

Table 22: CIFAR-10 Transfer learning benchmark tests with varying budget.

| Attack | Budget | Success Rates (%) | |
| --- | --- | --- | --- |
| | | White-box | Black-box |
| FC | 50 | $23.0 \pm 4.21$ | $7.0 \pm 1.80$ |
| | 100 | $59.0 \pm 4.92$ | $5.0 \pm 1.56$ |
| | 250 | $93.0 \pm 2.55$ | $17.0 \pm 3.76$ |
| FC-Ens. | 50 | $17.0 \pm 3.76$ | $19.5 \pm 2.80$ |
| | 100 | $40.0 \pm 4.90$ | $33.0 \pm 3.34$ |
| | 250 | $79.0 \pm 4.07$ | $69.0 \pm 4.62$ |
| CP | 50 | $24.0 \pm 4.27$ | $8.0 \pm 1.92$ |
| | 100 | $38.0 \pm 4.85$ | $2.5 \pm 5.00$ |
| | 250 | $49.0 \pm 5.00$ | $5.0 \pm 1.54$ |
| CP-Ens. | 50 | $22.0 \pm 4.14$ | $22.0 \pm 2.93$ |
| | 100 | $33.0 \pm 4.70$ | $28.0 \pm 3.17$ |
| | 250 | $47.0 \pm 4.99$ | $37.0 \pm 4.83$ |
| CLBD | 50 | $5.0 \pm 5.00$ | $3.0 \pm 1.20$ |
| | 100 | $2.0 \pm 5.00$ | $2.0 \pm 5.00$ |
| | 250 | $1.0 \pm 5.00$ | $2.0 \pm 5.00$ |
| HTBD | 50 | $7.0 \pm 2.55$ | $7.0 \pm 1.80$ |
| | 100 | $2.0 \pm 5.00$ | $6.5 \pm 1.74$ |
| | 250 | $10.0 \pm 3.00$ | $6.0 \pm 2.37$ |

### A.15 HOW MANY IMAGES

When scraping data from Google, the sources of images are diverse. If each source is responsible for very few images, then an adversary may have a difficult time poisoning a significant amount of data scraped by their victim. In order to investigate the diversity of sources, we query Google Images for each CIFAR-10 class label and measure how many images come from each source. Table 23 shows how many images the first through fifth most represented source are each responsible for in the first 100 search results for each class. Entries represent the number of images in a particular class coming from a particular source within the first 100 search results for that query. The first column represents the source responsible for the most images. The second column represents the source responsible for the second most images, etc. We find that sources generally are not highly dominant, and each source is responsible for few images. Poisoning methods which only perturb data from the target class may only be able to poison a very small percentage of the victim's total data, especially when the number of classes is high. For example, in a 1000-class problem like ImageNet, even if the attacker could poison $10\%$ of the target class, this would only represent $0.01\%$ of the total dataset. This percentage is far smaller than the percentages studied in the data poisoning literature.

Table 23: Google Images case study.

| | Number of Images | | | | |
|---|---|---|---|---|---|
| Search term | Source 1 | Source 2 | Source 3 | Source 4 | Source 5 |
| airplane | 7 | 5 | 5 | 3 | 3 |
| automobile | 9 | 7 | 6 | 6 | 3 |
| bird | 7 | 5 | 4 | 4 | 4 |
| cat | 8 | 8 | 5 | 4 | 4 |
| deer | 6 | 6 | 5 | 4 | 4 |
| dog | 14 | 9 | 6 | 4 | 3 |
| frog | 5 | 4 | 4 | 4 | 3 |
| horse | 9 | 5 | 4 | 3 | 3 |
| ship | 9 | 6 | 5 | 5 | 4 |
| truck | 9 | 6 | 6 | 5 | 4 |

## A.16 BENCHMARK RESULTS

We present complete benchmark results with confidence intervals in Tables 24, 25, and 26. All figures here are success rates of attacks reported as percentages.

Table 24: Complete CIFAR-10 transfer learning benchmark results.

| | **Transfer learning** | |
|---|---|---|
| Attack | WB | BB |
| FC | $16.0 \pm 3.67$ | $3.5 \pm 1.30$ |
| CP | $24.0 \pm 4.27$ | $4.5 \pm 1.47$ |
| BP | $85.0 \pm 3.57$ | $8.5 \pm 1.97$ |
| CLBD | $3.0 \pm 5.00$ | $3.5 \pm 1.30$ |
| HTBD | $2.0 \pm 5.00$ | $4.0 \pm 1.39$ |

Table 25: Complete CIFAR-10 from-scratch benchmark results.

| | **Training from scratch** | | | |
|---|---|---|---|---|
| Attack | ResNet-18 | MobileNetV2 | VGG11 | Average |
| FC | $0.0 \pm 5.00$ | $1.0 \pm 5.00$ | $3.0 \pm 5.00$ | $1.3 \pm 5.00$ |
| CP | $0.0 \pm 5.00$ | $1.0 \pm 5.00$ | $1.0 \pm 5.00$ | $0.7 \pm 5.00$ |
| BP | $3.0 \pm 5.00$ | $3.0 \pm 5.00$ | $1.0 \pm 5.00$ | $2.3 \pm 5.00$ |
| WiB | $45.0 \pm 4.97$ | $25.0 \pm 4.33$ | $8.0 \pm 2.71$ | $26.0 \pm 4.38$ |
| CLBD | $0.0 \pm 5.00$ | $1.0 \pm 5.00$ | $2.0 \pm 5.00$ | $1.0 \pm 5.00$ |
| HTBD | $0.0 \pm 5.00$ | $4.0 \pm 5.00$ | $1.0 \pm 5.00$ | $2.7 \pm 5.00$ |

Table 26: Complete TinyImageNet benchmark results.

| | **Transfer learning** | | **From scratch** |
|---|---|---|---|
| Attack | WB | BB | VGG16 |
| FC | $49.0 \pm 4.99$ | $2.0 \pm 5.00$ | $4.0 \pm 10.0$ |
| CP | $14.0 \pm 3.47$ | $1.0 \pm 5.00$ | $0.0 \pm 10.0$ |
| BP | $100.0 \pm 5.0$ | $10.5 \pm 3.06$ | $44.0 \pm 9.93$ |
| WiB | - | - | $32.0 \pm 9.33$ |
| CLBD | $3.0 \pm 5.0$ | $1.0 \pm 5.0$ | $0.0 \pm 10.0$ |
| HTBD | $3.0 \pm 5.0$ | $0.5 \pm 5.0$ | $0.0 \pm 10.0$ |

