# OpenReview forum: "Just How Toxic is Data Poisoning?  A Benchmark for Backdoor and Data Poisoning Attacks"
_ICLR.cc/2021/Conference — Reject_

### Official Review · AnonReviewer1 · 2020-10-26
**A good paper with needed clarity in a convoluted sub field**

**Rating:** 8
**Confidence:** 4

**Review:**

This paper discusses many issues in the data poisoning literature. They call into question the real world applicability of data poisoning attacks and discuss the short coming in the literature. They present several issues in the comparison of such attacks. Notably they show that many attributes of a data poisoning attack assumed to be irrelevant to the success of the attack are in fact statistically significant, such as a dataset size. Finally they present a new benchmark which attempts to standardize the comparison between data poisoning attacks.

I think this paper will benefit the data poisoning literature by illustrating its short comings and presenting a path forward.

----------------------------
Suggestions:

in the '“Clean” attacks are sometimes dirty ' section the authors present two examples, but a human study could be set up in order to establish how many 'clean' attacks are actually identifiable. A small Mechanical turk study would greatly increase the strength of this argument and should not be financially prohibitive.

The section on 'Performance is not invariant to dataset size ' convinces the reader that this attribute thought to be irrelevant to the success of the attack is relevant, but proposes no mechanism by which the data set size could effect the attack. A discussion section attempting to provide an answer to this question would be interesting, but is perhaps out of scope.

The section on data augmentation does not mention if the augmentations are also applied to the poisoned inputs. If the clean data is augmented but the poisoned data is not, it seems expected that the success of the attack would drop.

---

> ### Author Response · Authors · 2020-11-23
> **Rebuttal**
>
> Thank you for the positive feedback.  We agree that a human study might provide deeper insight regarding clean vs. dirty images, and we are now taking your suggestion and looking into this possibility.
>
> Regarding the scaling of poisoning methods with dataset size, we agree that this would be an interesting question to answer, and we have explored this topic, but we did not want to include any speculation until we have strong evidence.
>
> The poisoned inputs are also augmented, and we have updated the paper to be more clear.

---

### Official Review · AnonReviewer2 · 2020-10-28
**Solid methodology**

**Rating:** 7
**Confidence:** 4

**Review:**

The authors study a number of existing data poisoning attacks, ablating different design choices of these attacks and evaluating them on a common benchmark. They find that many of these attacks are quite brittle to changes in their original experimental evaluation and fail to generalize to more realistic/challenging scenarios. Then, they propose a unified benchmarks for evaluating these attacks.

Overall, I found the paper quite interesting. As far as I can tell, the experimental methodology is solid and clearly demonstrates the brittleness of many of these attacks. Similarly, the benchmark proposed presents a reasonable and realistic setting to evaluate attacks on. I believe that such unified benchmarks are important for making progress in such an actively growing field. While early attack approaches serve as proof of concept for different vulnerabilities, understanding when these attacks fail and where should one focus to improve them is necessary to make progress.

Other comments (not affecting score):
- S2, 1st paragraph: Early backdoor attacks were not clean-label because they also explicitly flip the label of the poisoned examples.
- S2, 3rd paragraph: Backdoor attacks cause the victim to misclassify everything _except_ the target class (adding the trigger causes the model to predict the target class).
- The CLBD attack does examine a modified trigger that works in the presence of data augmentation.
- Is there some intuition as to why the success rate of CP goes down when adding more training samples? If I understand correctly, the success of CLBD increases since the overall accuracy of the model improves with additional data (the model is more likely to rely on the trigger to perform well on the poisoned examples which, after all are inconsistent with the clean classifier).

====== POST-RESPONSE UPDATE ======
After reading the authors' responses and the rest of the reviews, I still stand by my original score.

That being said, I do agree with the other reviewers that the framing of certain discussion points could be improved to avoid misconceptions. Specifically:
- There are places where the narrative feels accusatory towards earlier attacks (e.g., words like "flaws").
- It would be good to explicitly state the scope of the paper relatively early (e.g., "clean-label attacks on deep learning for images") and explain this choice.
- Clarify that this is not meant to be the ultimate poisoning benchmark but rather a first step that allows us to make progress in terms of attack development and evaluation.

It would be great if these could be incorporated in the next revision of the manuscript.

---

> ### Author Response · Authors · 2020-11-23
> **Rebuttal**
>
> Thank you for the positive feedback. We have made the corrections to S2 in the latest draft, thank you for bringing those things to our attention. CLBD does include experiments with data augmentation, and it is noted with a check in Table 1. In our experiments, we chose to use the version of the trigger that does not include that modification since it is the predominantly studied method in the original work.
>
> The behavior shown in Figure 2 supports the claim that these attacks do not scale the same way.  Exactly why poisoning methods exhibit different scaling behavior is a difficult question and an interesting topic for future work.  We put some thought into this question, but we prefer to avoid speculation on this front without strong evidence.

---

### Official Review · AnonReviewer4 · 2020-11-05
**Attacks do not need to work in all settings**

**Rating:** 5
**Confidence:** 4

**Review:**

**Summary:** This paper studies 4 different previously proposed (clean label and targeted) poisoning attacks and compares them in a systematic way. The authors argue that the experimental setup in these papers are significantly different and comparing them with each other is hard. This incentivizes them to use a benchmark to test all in a unified way. They show that these 4 attacks are not robust to changing parameters of learning algorithm or the experimental setting. To fix this issue, the paper suggest a benchmark for testing poisoning attacks. Their benchmark is only for "clean label" and "targeted" poisoning attacks. They identify several issues with the experimental settings of previous poisoning attacks and suggest some guidelines on  for evaluating poisoning attacks.

**Evaluation:** I like the paper's analysis of different attacks and showing when they fail and more importantly, when they succeed. However, In my opinion, I don't find the findings of the paper in lack of robustness for the attacks surprising. These attacks are all proof of concepts that show a serious security concerns for real world applications. It is not surprising that none of them cannot succeed in all scenarios and in a robust manner. Attackers can always make tweaks to their algorithm for individual settings (adaptive attacks).

In general, I think designing adaptive benchmarks for attacks is not a very good idea. There might not exist a single attack that succeeds in all the scenarios in benchmark and this can give a false sense of security. Even if there is an attack that passes all the tests in the benchmark, one can try and find a scenario where the attack fails and add it to the benchmark. The authors might argue that the goal of the benchmark is not to show that the security threat is not real or that the attacks are weak, but to give a way to compare different attacks and provide insight for when attacks are (un)successful. I think this goal would be much more valuable as it poses a lot of open question for future research, but the current tone of the paper seems not to focus on that.

Comment to authors:
-The paper starts great by talking about the relevance of poisoning attacks in real world applications, but then the benchmarks are only for image classification. I think the focus on image classification is disproportionate to real world applications of machine learning.
- Any particular reason why you don't consider indiscriminative attacks?
-In the description of attacks it is not specified how the labels of the poison points are selected.
-In the blackbox setting, does the attacker know the clean training set?

---

> ### Author Response · Authors · 2020-11-23
> **Rebuttal**
>
> Thank you for taking the time to read the paper and for the constructive feedback. While the brittle nature of these attacks may not be surprising, we find that the literature to date skirts this phenomenon by testing methods only in settings in which they perform well, precisely creating the need for a benchmark and for publishing negative results.
>
> We agree that the goal of the benchmark is to compare methodologies and provide results that lend insight. We have made this clearer in our current draft.
>
> We agree that a benchmark for indiscriminative attacks would also benefit the community.  However, these attacks have not been successful for deep learning, and there thus has been very little recent literature on indiscriminative attacks for neural networks.  Thus, we chose to focus on targeted attacks since there is much more recent literature.
>
> Regarding selection of target labels, we choose these randomly.
>
> In the black-box setting of our benchmark, the attacker is allowed to know the clean training data.  However, none of the attacks we benchmark use the training data which they are not modifying.  We acknowledge that this may be realistic in some settings and unrealistic in others.

---

### Official Review · AnonReviewer3 · 2020-11-05
**Review for Just How Toxic is Data Poisoning? A Benchmark for Backdoor and Data Poisoning Attacks**

**Rating:** 4
**Confidence:** 5

**Review:**

Reject

This paper consists of two main parts, essentially inspired from the argumentation of the Baconian method. In particular, the first part criticizes current work in the area of data poisoning attacks (pars destruens), while the second part tries to overcome the limits described in the first half of the paper (pars construens).

In their pars destruens, the authors heavily criticize the lack of standardized evaluations across different papers, claiming that: "inconsistent and perfunctory experimentation has rendered performance evaluations and comparisons misleading". The authors then continue by highlighting the inconsistencies observed across different papers and summarize them in Table 1.

First, I think that this criticism is not fully justified. None of the considered papers had the goal of proposing a common benchmark evaluation. They were all proposing different backdoor attacks, under different scenarios. Hence, it is unfair to claim that their evaluations are inconsistent. Of course they are. Every paper considers a more or less different experimental setups according to their hypotheses, to validate or reject them. And this is true also for many other different research topics and areas. I believe that the authors should revise the presentation of their paper, acknowledging that a benchmark methodology is lacking and it is required, but without blaming the others because they did not develop it. Their goal was different.

Second, this work does not consider the whole family of poisoning attacks, but only "targeted" (or better, integrity) ones - this includes backdoor attacks or attacks aimed to misclassify only specific test samples, but not poisoning attacks that aim to indiscriminately increase the test error (i.e., availability / denial-of-service attacks). In addition, the whole paper only considers deep neural networks, and not other models. This should be thus clarified since the beginning in the paper, and better reflected also in the title. A clearer taxonomy of the whole family of data poisoning attacks should also be reported (e.g., in the form of a table to help understand the existing different types of threat in the area of data poisoning).
I recommend the authors to refer to these papers that may help categorizing data poisoning attacks (and systematize nomenclature):
- https://arxiv.org/abs/1910.03137
- https://arxiv.org/abs/1712.03141
- https://dl.acm.org/doi/10.1145/2046684.2046692


The main inconsistencies/issues identified by the authors in the evaluation of backdoor attacks are delineated in Sect. 4:
1. Training without SGD or data augmentation;
2. Victim architecture matters;
3. "Clean" attacks are sometimes dirty;
4. Proper transfer learning is less vulnerable;
5. Performance is not invariant to dataset size;
6. Black-box performance is low;
7. Small sample sizes and non-random targets;
8. Attacks are highly specific to the target image;
9. Backdoor success depends on patch size.

These 9 causes, according to the authors of this work, hinder the impact of the 4 backdoor attacks (FC, CP, CLBD, HTBD) considered in this paper.
I am quite convinced that in some specific cases, as the ones identified in Sect. 4, the attacks may fail, and I agree with the arguments posed by the authors in this section.

I am only concerned by Issue no. 3 about the need of "clean-label" attacks in realistic settings. This is a common criticism/misconception also related to adversarial examples with imperceptible perturbations.
Why the perturbation should be required to be small? Are there practical scenarios where humans are going to observe the samples and be trained to detect that these samples are "dangerous"?
As the authors of this work seem to be quite concerned on the realism of these attacks, this point should be better discussed, as well as the need of considering the perturbation model to be l-inf with size 8/255 (or anyway fixed). In this respect, note that one more pertinent motivation for requiring small perturbations may be the detectability of the attack by automatic tools (rather than imperceptibility to the human eye); see, e.g., the discussion in https://arxiv.org/abs/1802.07295 and consider expanding the paper to discuss this issue.

A final comment for the part destruens is that, eventually, it is not well systematized. Besides the 9 issues delineated above, a clear systematization/taxonomy of the potential issues is lacking. For example, issues can be related to the model (architecture), training algorithm (SGD/Adam, etc.), training hyperparameters, etc. Unfortunately, this step is lacking in the paper.
And, as we will see, this impacts the development of a proper evaluation framework.

After their part destruens, in Section 5, the authors move to the part construens of their argumentation, in which they propose a standardized benchmark for evaluation of clean-label and hidden trigger backdoor attacks. Again, I agree that providing a benchmark to assess poisoning attack effectiveness is a valuable contribution, and the authors have done a good job highlighting the factors which may impact the performance of these attacks.

However, I am also concerned about the proposed benchmarking framework.
In particular, as the authors have shown, factors such as the type of the target model, the training dataset size, and the size of the perturbation that the attacker can inject into the poisoning samples substantially impact the attack effectiveness. However, in the proposed framework, those factors assume a single value that may unreasonably favor an approach rather than another.
When a factor has a substantial impact on the results, it is recommended to analyze the performance when that factor assumes different values, as it is usually done for the size of the perturbation that an attacker can add to evasion samples (see, e.g., http://arxiv.org/abs/1902.06705).
More generally, a clear evaluation procedure or methodology is neither discussed nor provided, and this stems from the fact that a clear systematization of the causes of failure is lacking in the previous part of the paper.
If we identify, e.g., that the model architecture, the training algorithm and the perturbation size are all affecting the attack impact, a proper evaluation framework for attacks (and defenses) should then consider variants of all these factors, which means:
- testing the attack/defense on different models;
- (for each model) testing with different training algorithms;
- (for each model, training algorithm) testing with different perturbation sizes.
This would indeed give a much more detailed understanding of how an attack/defense performs w.r.t. previously-proposed or existing ones.

To conclude, I mostly liked the idea presented in the paper, but a much higher level of systematization is required to propose a comprehensive framework for evaluation of poisoning attacks and defenses, as well as clarifying that the scope of the framework is also restricted to backdoor/integrity attacks on DNNs. I would anyway encourage the authors to continue working on this benchmark to make it more systematic, fairer and inclusive.

---

> ### Author Response · Authors · 2020-11-23
> **Rebuttal**
>
> We thank the reviewer for the time and thorough review. We address the concerns in order.
>
> (a) We agree that previous papers that introduce attacks are not responsible for benchmark development. Our demonstration of inconsistencies is meant to motivate the need for a benchmark.  We feel that it is useful to know not only scenarios under which an attack is effective but also the scenarios under which it is NOT effective.  Additionally, our experiments highlight important factors for ablation studies in future work on data poisoning attacks (e.g. it’s useful to know how they interact with commonly used optimizers and data augmentation).
>
> (b) Thank you for suggesting that we contextualize our work amongst broader applications of data poisoning.  We have updated our current version to reflect that.  With that in mind, we chose to construct a benchmark for attacks on deep image classifiers as the popularity of image classification within recent (and likely future) poisoning literature dwarfs that of other popular applications such as recommender systems (Li et al. 2016, Fang et al. 2018, Hu et al. 2019) and speech recognition (Aghakhani et al. 2020).  Moreover, we feel that benchmarking is less important for earlier work on linear models given that provably optimal attacks are well-known, and ablation factors like optimizers have no influence as the loss is convex.  We feel that our work well-systematizes the area on which we focus.
>
> (c) We agree with you that poisons being undetectable by automated tools may be more important in scenarios in which victims are aware of the poisoning threat and try to mitigate it.  We have taken your suggestion and added a discussion of this in our latest draft.  We incorporated the “clean-label” condition into our benchmark and ablations as this is the explicit goal of many attacks (Turner 2018, Zhu 2019, Aghakhani ‎2020, Shafahi 2018, Saha 2019, Geiping 2020).  Works do exist on adaptive poisoning attacks (e.g. Koh 2018) which avoid detection or mitigation, but this domain is still relatively unexplored, and we thus avoid it in our benchmark.
>
> (d) As you say, it would be useful to see all of these ablations for every poisoning method proposed in the future, and we hope that our own ablations inspire this.  However, for the benchmark to be approachable and computationally feasible for practitioners, we choose to set each ablation factor to a fixed value.  In the paper and in the appendix, we justify these choices. For example, see Table 20 in A.13 and Table 23 in A.15.

---

> ### Comment · AnonReviewer3 · 2020-11-25
> **Response to the authors' rebuttal**
>
> I'd like to thank the authors for their response and clarifications. I think there's still need to clarify many points in their work before publication.
>
> First, clarity is lacking in the definition of the threat models for poisoning attacks (and accordingly on what should be tested by the proposed framework).
>
> (a) Backdoor attacks are targeted poisoning attacks. They assume that the attacker controls the design phase and the training process, and releases a backdoored model (which then the defender re-uses possibly with fine tuning). In this setting, clean-label attacks do not make sense (as the attacker controls the training labels too).
>
> (b) Clean-label poisoning attacks are still targeted, but assume a different setup. Here the attacker can only inject poisoning samples into the training set of the defender but does neither control the training process nor the training labels. This is why clean-label attack samples are required (the threat model assumes that someone else is going to label these images).
>
> (c) Poisoning availability/indiscriminate attacks aim to cause a denial of service, maximizing the classification error. They are completely different from the attacks in (a) and (b), even though the same bilevel optimization problem may be adjusted to craft optimal targeted attacks (by selecting a target subset of the validation points in the outer objective).
>
> Second, previous work on (c) was not restricted to linear models (nonlinear SVMs were considered, as well as linear models trained on nonlinear representations learned with deep neural networks). Moreover, we should look at the convexity of the outer loss in the bilevel problem and not simply at that of the training problem of the classifier (w.r.t. the input data x). In this case, I do not think that convexity is guaranteed even if the underlying model is linear. In addition, the hypergradients computed to optimize the outer loss in the bilevel optimization process are often really noisy. Hence, the sentence "ablation factors like optimizers have no influence as the loss is convex" is simply wrong.
>
> Overall, I think that this work has potential, but in its current form I found that more clarity is required and the proposed framework is not sufficiently detailed and thorough to provide a proper benchmark for data poisoning attacks.

---

### Decision · Program_Chairs · 2021-01-07
**Final Decision**

**Decision:**

Reject

**Comment:**

This paper generated significant discussion and division amongst the reviewers. On the positive side, some reviewers enjoyed both contributions, feeling the further empirical investigation of existing attacks to be interesting, and the creation of a benchmark to be very useful. On the negative side, no new positive results were proposed, criticism of previous attacks were considered to be unjust, the focus was somewhat narrow, and a benchmark could plausibly be misleading and detrimental.

Given the highly competitive nature of ICLR and the many other excellent submissions, the committee was unable to accept the paper at this time. Below are some suggestions for future submissions.

The content of the paper is generally fine, as long as the caveats and the "tone" are appropriate: we would hope to not mislead potential readers.
Here are some (strong) recommendations:
- Previous works were proof of concept attacks, and the authors should be careful to not frame them as being "broken" -- they perhaps were not meant to be robust to these modifications.
- The scope is somewhat narrow. There should be some explicitly statement and justification of the scope, and what in particular is *not* covered by the investigation.
- Importantly, a single benchmark can't be a unique gold standard, for many reasons discussed by reviewers. Please state these caveats clearly and prominently in the paper and/or code release, as otherwise the presence of a benchmark could do more harm than good. In particular, Reviewer 4 brought up the following philosophical concern with a benchmark, which I believe is quite reasonable, and I reproduce verbatim. The authors should try to address this in the next version: "This kind of benchmark can push the research to a wrong direction. In my view, the point of attacks are to create an alarm for using machine learning in critical applications. Developing these benchmarks would push the competition in the direction of making existing attacks "better" (whatever "better" means in the benchmark) instead of focusing on designing defense techniques or showing the severity of attacks in other situations. This benchmark could also have a bad effect on future attacks (attacks that want to show a new threat, not the one that try to improve the performance of clean label targeted poisoning attacks on deep neural nets) to gain attention from community as they probably will not pass all the criteria of this benchmark."

As another comment (I believe mentioned by other reviewers), it would be nice if all the terms and settings were defined clearly and precisely. For a benchmarking paper, it is important that the reader can clearly understand the threat model, and what does and does not count as a valid attack.

Finally, many of the reviewers gave detailed comments and concerns. The authors should please note and discuss these concerns in future versions (or at least in a supplement or an arXiv version).